# AF-React study: atrial fibrillation management strategies in clinical practice—retrospective longitudinal study from real-world data in Northern Portugal

Susana Silva Pinto [iD] ,[1,2,3] Andreia Teixeira [iD] ,[1,2,4] Teresa S Henriques,[1,2] Hugo Monteiro,[5] Carlos Martins [iD] [1,2]

For numbered affiliations see end of article.

**Correspondence to**
Dr Susana Silva Pinto; susyapinto@gmail.com

## ABSTRACT

**Objectives** To determine the prevalence of atrial fibrillation (AF) and to assess how these patients are being cared for: what anticoagulants are being prescribed and are they being prescribed as recommended?

**Design** Retrospective longitudinal study.

**Setting** This study was conducted in the Regional Health Administration of Northern Portugal.

**Participants** This study used a database that included 63526 patients with code K78 of the International Classification of Primary Care between January 2016 and December 2018.

**Results** The prevalence of AF among adults over 40 years in the northern region of Portugal was 2.3% in 2016, 2.8% in 2017 and 3% in 2018. From a total of 63 526 patients, 95.8% had an indication to receive anticoagulation therapy. Of these, 44 326 (72.9%) are being treated with anticoagulants: 17 936 (40.5%) were prescribed vitamin K antagonists (VKAs) and 26 390 (59.5%) were prescribed non-VKA anticoagulants. On the other hand, 2688 patients of the total (4.2%) had no indication to receive anticoagulation therapy. Of these 2688 patients, 1100 (40.9%) were receiving anticoagulants.

**Conclusions** The prevalence of AF is 3%. Here, we report evidence of both undertreatment and overtreatment. Although having an indication, a considerable proportion of patients (27.1%) are not anticoagulated, and among patients with AF without an indication to receive anticoagulation therapy, a considerable proportion (40.9%) are receiving anticoagulants. The AF-React study brings extremely relevant conclusions to Portugal and follows real-world studies in patients with AF in Europe, presenting some data not yet studied.

## BACKGROUND

Atrial fibrillation (AF) is the most common type of sustained cardiac arrhythmia.[1] The clinical relevance of AF is associated with the loss of effective atrial contractility, resulting in deficient emptying of the left atrial appendage. This process increases the risk of thrombus formation and thromboembolic events.[1]

Due to the ageing of the population, the number of people affected by AF worldwide is projected to exceed 12 million by 2050.[2] In Portugal in 2010, the FAMA study[3] verified an overall prevalence of AF of 2.5% in a population sample older than 40 years, with a significant increase after the age of 70 years.[3]

In Portugal, despite a considerable reduction in ischaemic stroke mortality below 70 years (39% reduction) in 2015 compared with 2011, stroke remains the main cause of mortality related to vascular diseases.[4] Patients with AF have a fivefold increased risk of stroke.[5] This arrhythmia is currently responsible for about 15% of stroke cases.[3] Stroke is a major complication associated

### Strengths and limitations of this study

► The main limitation of this retrospective study is that the atrial fibrillation (AF) assessment is based on data coded in the clinical process in primary healthcare.

► There might be more patients with AF in the northern region of Portugal, but they are not coded at primary healthcare and, therefore, were not included in this study.

► The lack of International Classification of Primary Care-2 coding for problems such as mechanical prosthesis valve and rheumatic moderate or severe mitral stenosis might be another limitation.

► Few of the 63 526 patients will be diagnosed with atrial flutter; however, for patients with atrial flutter, anticoagulant therapy is recommended according to the same risk profile used for AF.

► The fact that it is not known when treatment started for patients already diagnosed with AF before 2016 may be a limitation for interpreting the data.

with AF, which contributes to the morbidity and mortality associated with the disease.[6]

Previous studies have reported that oral anticoagulation with vitamin K antagonists (VKAs), such as warfarin, reduces stroke and mortality in patients with AF.[7] The non-VKA oral anticoagulants (NOAC)—dabigatran, rivaroxaban, apixaban, edoxaban—have been shown to be superior to warfarin in the prevention of thromboembolic events in patients with non-valvular AF, providing increased safety and an overall reduction in the number of cases of bleeding events.[8–11] The cost and morbidity associated with stroke and intracranial haemorrhage are high, so the lower prevalence of these events with NOACs, compared with warfarin, is one factor that might help this drug to become cost-effective relative to warfarin.[12] Furthermore, according to the Portuguese Directorate-General of Health, the reduction in ischaemic stroke mortality below 70 years was due to the introduction of NOAC into clinical practice as an antithrombotic therapy for AF.[4]

In a cross-sectional study of patients with AF carried out in eight Family Health Units of Northern Portugal in 2015 (the FATA study),[13] only 56.8% of patients with AF were prescribed adequate oral anticoagulation according to the recommendations of the European Society of Cardiology (ESC) Guidelines in 2010. In the SAFIRA study (2018),[14] 56.3% of patients with AF were not anticoagulated.

There are no randomised controlled trials (RCTs) directly comparing NOACs to each other (or to placebo). All RCTs compared an NOAC to warfarin. For example, the RE-LY,[8] ROCKET-AF,[9] ARISTOLETE[10] and ENGAGE AF-TIMI 48[11] studies included 18 113, 14 264, 18 201 and 21 105 patients with AF, respectively. In fact, these trials were crucial for the efficacy and safety evaluation of these drugs and their approval. However, real-world studies might contribute additional relevant evidence because they include epidemiological data and drug prescribing patterns for a larger number of patients in the context of daily clinical practice. In Portugal, there are no real-world studies with a large database of oral anticoagulation in patients with AF.

In Europe, there are some studies with a large database. A study involving databases from six European countries concluded that, overall, apixaban and rivaroxaban increased their use during the study period while dabigatran decreased and there was variability in patient characteristics such as comorbidities, potentially interacting drugs, and dose adjustment.[15] Komen et al[16] investigated the influence of patient characteristics such as age, stroke and bleeding risks on decisions for antithrombotic treatment in patients with AF and concluded that apixaban was favoured for elderly and high-risk patients, whereas dabigatran was used in lower risk patients.[16] Mueller et al[17] compared the clinical effectiveness and safety of NOACs in patients with AF in routine clinical practice. All NOACs were similarly effective at preventing strokes and systemic embolisms, while patients being treated with rivaroxaban exhibited the highest bleeding risks.[17]

The AF-React study aims to determine the prevalence of AF in the northern region of Portugal and to assess how patients with AF are being cared for: what anticoagulants are being prescribed and are they being prescribed as recommended?

## METHODS

The Portuguese National Health Service is a universal health coverage system and is administratively divided into five regions: North, Centre, Lisbon and Tagus Valley, Alentejo and Algarve. The entirety of the mainland Portuguese population is enrolled in one of these administrative health regions. Each regional health administration is responsible for providing primary and secondary healthcare to the population living in its geographic area. This retrospective longitudinal study was conducted using data from the Regional Health Administration database of Northern Portugal.

The Department of Studies and Planning of the Regional Health Administration of Northern Portugal built a database containing the following data for the years 2016, 2017 and 2018: population enrolled in the Regional Health Administration of Northern Portugal, number of patients diagnosed with other International Classification of Primary Care (ICPC)-2 coded health problems,[18] and last record available of body mass index, cardiovascular (CV) risk score and glomerular filtration rate (GFR). In fact, this is one-off data (once a year) for all variables (age, gender, professional situation, ICPC-2 coded health problems, body mass index, CV risk score and GFR) except the anticoagulants prescription history, for which we had longitudinal information with the date of each prescription.

To identify patients with AF, the ICPC-2 code for AF, K78, was used.[18] Therefore, we included all adults (age ≥18 years) with the code K78 between January and December of the years 2016, 2017 and 2018. For each year, either previous AF diagnosed patients or new AF diagnosed patients were included, so the database comprises both incident and prevalent patients.

The prevalence of AF was calculated for the population enrolled in the Regional Health Administration of Northern Portugal in 2016, 2017, and 2018.

The CV risk was defined in four stages based on SCORE (Systematic Coronary Risk Evaluation), as low, moderate, high and very high, if less than 1%, between 1% and 5%, between 5% and 10%, and greater than or equal to 10%, respectively, for patients between 40 and 65 years old.

The stages of chronic kidney disease (CKD) were defined based on the GFR calculated by the equation of *Cockcroft-Gault*: stage 1, GFR ≥90 mL/min; stage 2, GFR 90–60 mL/min; stage 3, GFR 60–30 mL/min; stage 4, GFR (30–15) mL/min and stage 5, GFR <15 mL/min. The high and very high stages of CV risk were based on the codification of health problems by ICPC-2 and the patient's GFR registered in the clinical process.

The body mass index was defined in six categories: <18.5 kg/m² —underweight; 18–25 kg/m²—normal; 25–30 kg/m²—overweight; 30–35 kg/m²—obese class I; 35–40 kg/m²—obese class II and ≥40 kg/m²—obese class III.

The $CHA_2DS_2$-VASc score was calculated according to the ESC guidelines in 2016 for AF,[19] as follows by ICPC-2 coding[18]: C-congestive heart failure (K77): 1 point; H-hypertension (K86 or K87): 1 point; $A_2$-age >75 years or older: 2 points; D-diabetes mellitus (T89 or T90): 1 point; $S_2$-stroke (K89, K90 or K91): 2 points; V-vascular disease (K75 or K92): 1 point; A-age 65–74 years: 1 point and Sc-sex category (female): 1 point. According to the 2016 ESC guidelines,[19] we considered a $CHA_2DS_2$-VASc score ≥1 point in a man and a $CHA_2DS_2$-VASc score ≥2 points in a female to be an indication of anticoagulation.

The categorical variables were described using absolute and relative frequencies, n (%). The normally distributed continuous variables were described by the mean and respective SD, mean±SD and by the minimum (min) and maximum (max) values. In the case of continuous variables not normally distributed, the data are presented by the median and respective IQR, Med $(Q_1; Q_3)$, where $Q_1$ is the first quartile and $Q_3$ is the third quartile. The normality was verified by observing the histograms.

The database was exported to Microsoft Excel 2016 and the statistical analysis was performed using SPSS Statistics 25.0 and software R. Data were obtained without aggregation at the individual level for general characterisation analysis.

The data processing was carried out through the Department of Studies and Planning of the Regional Health Administration of Northern Portugal (Ministry of Health, Portugal). The data were extracted from the server through an anonymised data processing and editing platform and delivered securely (and in accordance with legal regulations and due approval) to the principal investigator.

### Patient and public involvement
Patients and the public were not involved in the development of the research question.

### RESULTS
A total of 63 526 patients were identified as having the diagnosis of atrial fibrillation/flutter (ICPC-2, K78 code) in the northern region of Portugal. The average age was 76.5±10.6 years (range from 18 to 107 years) and 53% were women. Table 1 provides a detailed characterisation of this patient group. Considering the 2 077 599 adults over 40 years enrolled in 2018 in the Regional Health Administration of Northern Portugal, the prevalence of AF was 3.044% (table 2). In 2017, among the 1 943 813 adults over 40 years enrolled in the Regional Health Administration of Northern Portugal, the prevalence of AF was 2.787%. In 2016, among the 2 024 390 adults over 40 years enrolled in the Regional Health Administration of Northern Portugal, the prevalence of AF was 2.295%.

**Table 1** Characterisation of the atrial fibrillation patients diagnosed in the Regional Health Administration of Northern Portugal (n=63 526)

| | |
|---|---|
| Gender, n (%) | |
| Female | 33 437 (52.6%) |
| Male | 30 089 (47.4%) |
| Age (years), mean±SD, min, max | 76.5±10.6, 18, 107 |
| Age groups, n (%) | |
| <40 years | 286 (0.5) |
| 40–49 years | 812 (1.3) |
| 50–59 years | 3286 (5.2) |
| 60–69 years | 10 089 (15.9) |
| 70–79 years | 20 602 (32.4) |
| ≥80 years | 28 451 (44.8) |
| Professional situation, n (%) | |
| Active | 13 553 (21.3) |
| Not active | 4957 (7.8) |
| Student | 54 (0.1) |
| Retired | 44 685 (70.3) |
| Unknown | 277 (0.5) |
| Body mass index, n (%) | 9448 (14.9%) |
| Body mass index*, Med $(Q_1; Q_3)$ | 29.1(26.2; 32.6) |
| Body mass index categories, n (%) | |
| Underweight | 29 (0.3) |
| Normal | 1272 (13.5) |
| Overweight | 4126 (43.7) |
| Obese class I | 2725 (28.8) |
| Obese class II | 1297 (13.7) |
| Glomerular filtration rate†, n (%) | |
| Value available | 52 207 (82.2%) |
| Missing record | 11 319 (17.8%) |
| Stage of chronic kidney disease | |
| 1 | 11 044 (21.2%) |
| 2 | 19 693 (37.7%) |
| 3 | 19 307 (37.0%) |
| 4 | 2110 (4.0%) |
| 5 | 53 (0.1%) |
| Cardiovascular risk for patients with age between 40 and 65 years (n=8985), n (%) | |
| Value available | 6531 (72.7%) |
| Missing record | 2454 (27.3%) |
| Classification of cardiovascular risk‡ (n=6531), n (%) | |
| Low | 650 (10.0%) |
| Moderate | 2631 (40.3%) |
| High | 194 (3.0%) |
| Very high | 3056 (46.7%) |
| $CHA_2DS_2$-VASc score, n (%) | |

Continued

**Table 1** Continued

| 0 | 1658 (2.6%) |
|---|---|
| 1 | 3967 (6.2%) |
| 2 | 8065 (12.7%) |
| 3 | 13 811 (21.7%) |
| 4 | 16 152 (25.4%) |
| 5 | 11 388 (17.9%) |
| 6 | 5520 (8.7%) |
| 7 | 2196 (3.5%) |
| 8 | 685 (1.1%) |
| 9 | 84 (0.1%) |

*The body mass index is defined under six categories: <18.5 kg/m² — underweight; 18–25 kg/m² — normal; 25–30 kg/m² — overweight; 30–35 kg/m² — obese class I; 35–40 kg/m² — obese class II; ≥40 kg/m² — obese class III.

†The stages of chronic kidney disease were defined based on the glomerular filtration rate (GFR) calculated by the equation of *Cockcroft-Gault*: stage 1, GFR ≥90 mL/min; stage 2, GFR 90–60 mL/min; stage 3, GFR 60–30) mL/min, stage 4, GFR 30–15 mL/min and stage 5, GFR <15 mL/min.

‡The cardiovascular risk was defined under four stages that were based on SCORE (Systematic Coronary Risk Evaluation), as low, moderate, high and very high, if less than 1%, between 1% and 5%, between 5% and 10%, and greater than or equal to 10%.

From a total of 63 526 patients, 60 838 (95.8%) had an indication to receive anticoagulation therapy: men with $CHA_2DS_2$-VASc score ≥1 and women with $CHA_2DS_2$-VASc score ≥2. Among these 60 838 patients, 16 512 (27.1%) were not being treated with anticoagulants, and 79% of those 16 512 had a $CHA_2DS_2$-VASc score of 3 or higher.

**Table 3** Medicated and non-medicated patients with and without a recommendation for anticoagulation

| Recommendation* | Medicated, n (%) | | Total, n |
|---|---|---|---|
| | Yes | No | |
| Yes | 44 326 (72.9) | 16 512 (27.1) | 60 838 |
| No | 1100 (40.9) | 1588 (59.1) | 2688 |
| Total | 45 426 (71.5) | 18 100 (28.5) | 63 526 |

*Oral anticoagulation was considered in a man with a $CHA_2DS_2$-VASc score ≥1 point and in a woman with a score ≥2 points, according to 2016 European Society of Cardiology Guidelines for the management of atrial fibrillation developed in collaboration with European Association of Cardio-Thoracic Surgery.

Among the 44 326 being treated with anticoagulants, 17 936 (40.5%) were prescribed VKAs (warfarin and acenocoumarol) and 26 390 (59.5%) with NOACs (dabigatran, rivaroxaban, apixaban and edoxaban). The most commonly prescribed drug in medicated patients was warfarin (31%), followed by rivaroxaban (23.6%), apixaban (20.3%), dabigatran (12.6%), acenocoumarol (9.6%) and edoxaban (2.9%).

The $CHA_2DS_2$-VASc score has the following distribution: 0 points (2.6%), 1–3 points (40.7%), 4–5 points (43.3%) and ≥6 points (13.4%). From the total of 30 089 men, 1658 (5.5%) had a $CHA_2DS_2$-VASc score of 0, and from a total of 33 437 women, 1030 (3.1%) had a $CHA_2DS_2$-VASc score of 1. This means that these 2688 (4.2%) patients of the total had no indication to receive anticoagulation therapy. Of these 2688 patients, 1100 were receiving anticoagulants (table 3).

**Table 2** Prevalence (%) of AF in the northern region of Portugal between 2016 and 2018

| | | <40 years | 40–49 years | 50–59 years | 60–69 years | 70–79 years | ≥80 years | Over 40 years |
|---|---|---|---|---|---|---|---|---|
| 2016 | Patients with AF (n) | 200 | 546 | 2184 | 7061 | 15 089 | 21 576 | 46 456 |
| | Population* (n) | 1 064 388 | 593 200 | 502 021 | 438 432 | 286 658 | 204 079 | 2 024 390 |
| | Prevalence (%) | 0.019 | 0.092 | 0.435 | 1.611 | 5.264 | 10.572 | 2.295 |
| 2017 | Patients with AF (n) | 239 | 678 | 2647 | 8442 | 17 665 | 24 746 | 54 178 |
| | Population† (n) | 952 914 | 506 129 | 527 554 | 433 722 | 258 239 | 218 169 | 1 943 813 |
| | Prevalence (%) | 0.025 | 0.134 | 0.502 | 1.946 | 6.841 | 11.343 | 2.787 |
| 2018 | Patients with AF (n) | 286 | 812 | 3286 | 10 089 | 20 602 | 28 451 | 63 240 |
| | Population‡ (n) | 957 693 | 588 852 | 530 424 | 432 439 | 307 823 | 218 061 | 2 077 599 |
| | Prevalence (%) | 0.030 | 0.138 | 0.620 | 2.333 | 6.693 | 13.047 | 3.044 |

*Population enrolled in primary healthcare in northern Portugal based on the age pyramid of Regional Health Administration of Northern Portugal in 2016.

†Population enrolled in primary healthcare in northern Portugal based on the age pyramid of Regional Health Administration of Northern Portugal in 2017.

‡Population enrolled in primary healthcare in northern Portugal based on the age pyramid of Regional Health Administration of Northern Portugal in 2018.

AF, atrial fibrillation.

**Table 4** Distribution of patients for CHA$_2$DS$_2$-VASc score value, gender and respective drugs

| CHA$_2$DS$_2$-VASc score | No OAC | NOAC group | | | | | VKA group | | |
|---|---|---|---|---|---|---|---|---|---|
| | Total | Total | Rivaroxaban | Apixaban | Dabigatran | Edoxaban | Total | Warfarin | Acenocoumarol |
| **Male** | | | | | | | | | |
| 0 | 990 (59.7) | 500 (30.2) | 248 (15.0) | 134 (8.1) | 88 (5.3) | 30 (1.8) | 168 (10.1) | 127 (7.7) | 41 (2.5) |
| 1 | 1069 (36.4) | 1313 (44.7) | 623 (21.2) | 344 (11.7) | 276 (9.4) | 70 (2.4) | 555 (18.9) | 426 (14.5) | 129 (4.4) |
| 2 | 1646 (28.6) | 2648 (46.0) | 1125 (19.5) | 801 (13.9) | 590 (10.2) | 132 (2.3) | 1468 (25.5) | 1099 (19.1) | 369 (6.4) |
| 3 | 2132 (25.3) | 3742 (44.5) | 1531 (18.2) | 1210 (14.4) | 823 (9.8) | 178 (2.1) | 2537 (30.2) | 1955 (23.2) | 582 (6.9) |
| 4 | 1549 (24.8) | 2572 (41.2) | 1037 (16.6) | 877 (14.0) | 547 (8.8) | 111 (1.8) | 2127 (34.0) | 1636 (26.2) | 491 (7.9) |
| 5 | 798 (25.0) | 1301 (40.7) | 471 (14.7) | 486 (15.2) | 286 (9.0) | 58 (1.8) | 1095 (34.3) | 813 (25.5) | 282 (8.8) |
| 6 | 377 (27.1) | 502 (36.1) | 183 (13.2) | 187 (13.4) | 102 (7.3) | 30 (2.2) | 512 (36.8) | 371 (26.7) | 141 (10.1) |
| 7 | 96 (23.1) | 164 (39.4) | 61 (14.7) | 58 (13.9) | 38 (9.1) | 7 (1.7) | 156 (37.5) | 116 (27.9) | 40 (9.6) |
| 8 | 19 (26.4) | 25 (34.7) | 12 (16.7) | 8 (11.1) | 2 (2.8) | 3 (4.2) | 28 (38.9) | 20 (27.8) | 8 (11.1) |
| Subtotal | 8676 (28.8) | 12 767 (42.4) | 5291 (17.6) | 4105 (13.6) | 2752 (9.1) | 619 (2.1) | 8646 (28.7) | 6563 (21.8) | 2083 (6.9) |
| **Female** | | | | | | | | | |
| 1 | 598 (58.1) | 272 (26.4) | 118 (11.5) | 88 (8.5) | 49 (4.8) | 17 (1.7) | 160 (15.5) | 125 (12.1) | 35 (3.4) |
| 2 | 757 (32.9) | 1004 (43.6) | 418 (18.2) | 345 (15.0) | 189 (8.2) | 52 (2.3) | 542 (23.5) | 432 (18.8) | 110 (4.8) |
| 3 | 1544 (28.6) | 2399 (44.4) | 951 (17.6) | 781 (14.5) | 515 (9.5) | 152 (2.8) | 1457 (27.0) | 1130 (20.9) | 327 (6.1) |
| 4 | 2555 (25.8) | 4460 (45.0) | 1708 (17.2) | 1565 (15.8) | 979 (9.9) | 208 (2.1) | 2889 (29.2) | 2195 (22.2) | 694 (7.0) |
| 5 | 2142 (26.1) | 3369 (41.3) | 1287 (15.7) | 1281 (15.6) | 694 (8.5) | 152 (1.9) | 2638 (32.4) | 2035 (24.8) | 603 (7.4) |
| 6 | 1117 (27.1) | 1666 (40.3) | 627 (15.2) | 623 (15.1) | 330 (8.0) | 86 (2.1) | 1346 (32.6) | 1024 (24.8) | 322 (7.8) |
| 7 | 525 (29.5) | 722 (40.6) | 232 (13.0) | 312 (17.5) | 150 (8.4) | 28 (1.6) | 533 (29.9) | 398 (22.4) | 135 (7.6) |
| 8 | 161 (26.3) | 257 (41.9) | 85 (13.9) | 121 (19.7) | 43 (7.0) | 8 (1.3) | 195 (31.8) | 148 (24.1) | 47 (7.7) |
| 9 | 25 (29.8) | 33 (39.3) | 11 (13.1) | 14 (16.7) | 6 (7.1) | 2 (2.4) | 26 (31.0) | 23 (27.4) | 3 (3.6) |
| Subtotal | 9424 (28.2) | 14 227 (42.5) | 5437 (16.3) | 5130 (15.3) | 2955 (8.8) | 705 (2.1) | 9786 (29.3) | 7510 (22.5) | 2276 (6.8) |
| Total | 18 100 (28.5) | 26 994 (42.5) | 10 728 (16.9) | 9235 (14.5) | 5707 (9.0) | 1324 (2.1) | 18 432 (29.0) | 14 073 (22.2) | 4359 (6.9) |

NOAC, non-vitamin K antagonist oral anticoagulants; OAC, oral anticoagulation; VKA, vitamin K antagonists.

The distributions of the different drugs by CHA$_2$DS$_2$-VASc score and gender are shown in table 4. For men, rivaroxaban (15.0%) and apixaban (8.1%) were more often prescribed than warfarin (7.7%) in the group with a CHA$_2$DS$_2$-VASc score of 0. In particular, apixaban was more prescribed than rivaroxaban in groups with

**Table 5**  Distribution of patients for chronic kidney disease' (CKD) stages by respective drugs

| Stages CKD | Drugs, n (%) | | | | | |
|---|---|---|---|---|---|---|
|  | Rivaroxaban | Apixaban | Dabigatran | Edoxaban | Warfarin | Acenocoumarol |
| 1 | 2164 (27.5) | 1531 (19.4) | 1081 (13.7) | 279 (3.5) | 2133 (27.1) | 691 (8.8) |
| 2 | 3673 (24.7) | 2944 (19.8 | 1980 (13.3) | 434 (2.9) | 4428 (29.8) | 1386 (9.3) |
| 3 | 3184 (22.3) | 3139 (22.0) | 1649 (11.6) | 394 (2.8) | 4504 (31.6) | 1393 (9.8) |
| 4 | 202 (14.8) | 338 (24.7) | 86 (6.3) | 28 (2.0) | 557 (40.7) | 157 (11.5) |
| 5 | 4 (11.1) | 11 (30.6) | 1 (2.8) | 1 (2.8) | 17 (47.2) | 2 (5.6) |
| Total | 9227 (24) | 7963 (20.7) | 4797 (12.5) | 1136 (3.0) | 11 639 (30.3) | 3629 (9.5) |

CHA$_2$DS$_2$-VASc scores of 5 or 6 (15.2% vs 14.7% and 13.4% vs 13.2%, respectively). For women, apixaban was more prescribed than rivaroxaban in the groups with CHA$_2$DS$_2$-VASc scores of 7, 8 or 9 (17.5% vs 19.7%, 16.7% vs 13.0% and 13.9% vs 13.1%, respectively).

The CV risk was available for 6531 (72.7%) of the patients between 40 and 65 years old. The CV risk values, for patients in that age group, were distributed as follows: low—1.0%, moderate—40.3%, high—3.0% and very high—46.8%.

The GFR was available for 52 207 patients (82.2%) and followed this distribution: stage 1 (21.2%); stage 2 (37.7%); stage 3 (37.0%); stage 4 (4.0%) and stage 5 (0.1%). The VKAs (warfarin and acenocoumarol) were more prescribed than NOACs (dabigatran, rivaroxaban, apixaban and edoxaban) in stages 4 and 5. The distribution of patients for CKD stages by respective drugs is represented in table 5.

Of 63 526 patients with AF, between 2016 and 2018, 17936 (28.2%) were always prescribed VKA and 21 854 (34.4%) were always prescribed NOAC. In particular, 4133 patients (6.5% of the total) had a single recorded prescription for any anticoagulant: warfarin—1496 (36.2%), rivaroxaban—882 (21.34%), apixaban—815 (19.71%), acenocoumarol—397 (9.61%), dabigatran—370 (8.95%) and edoxaban—173 (4.19%). Furthermore, 3508 (5.5%) were prescribed VKA and switched to NOAC, 1504 (2.4%) were prescribed NOAC and switched to another NOAC, and 305 (0.5%) were prescribed NOAC and switched to VKA. The following patients were always prescribed the same NOAC: rivaroxaban—8801 (22.18%), apixaban—7052 (17.78%), dabigatran—5219 (13.16%) and edoxaban—782 (1.97%).

The prevalence of each of the comorbidities studied in patients with AF is represented in table 6. We found that the most prevalent comorbidities associated with AF were hypertension (77.2%), dyslipidaemia (52.1%), diabetes (28.5%) and heart failure (27.6%).

## DISCUSSION
### Summary
In the AF-React study, the prevalence of AF was 3% in 2018 in population aged 40 years or older. A considerable proportion of patients, with an indication for anticoagulation therapy, are not anticoagulated (27.1%), and among patients with AF without indication to receive anticoagulation therapy, a considerable proportion (40.9%) are receiving anticoagulants.

### Comparison with existing literature
The prevalence of AF found in this study is similar to that of the FAMA study[3] (3% vs 2.5%), and higher than that of

**Table 6**  Comorbidities by the International Classification of Primary Care—2nd edition

| Comorbidities | n (%) |
|---|---|
| Chronic alcohol abuse (P15) | 2392 (3.8) |
| Tobacco abuse (P17) | 2880 (4.5) |
| Lipid disorder (T93) | 33 095 (52.1) |
| Atherosclerosis/PVD (K92) | 4617 (7.3) |
| Diabetes insulin dependente (T89) | 1551 (2.4) |
| Diabetes non-insulin dependente (T90) | 16 553 (26.1) |
| Coronary heart disease ischaemic heart disease w. angina (K74) | 3370 (5.3) |
| Ischaemic heart disease w/o angina (K76) | 3185 (5.0) |
| Acute myocardial infarction (K75) | 2450 (3.9) |
| Hypertension complicated (K87) | 17 598 (27.7) |
| Hypertension uncomplicated (K86) | 31 466 (49.5) |
| Heart failure (K77) | 17 530 (27.6) |

the FATA study[13] (3% vs 1.3%). The FAMA study[3] used a representative sample of the Portuguese population over 40 years old to study the prevalence of AF in 2010. The AF disease was determined using a 12-lead ECG. The FATA study[13] is an observational cross-sectional study from 2014 that included all patients aged 30 or above diagnosed with AF, enrolled in one of the eight Family Health Units of a northern region of Portugal. The higher values found in our study, as compared with the FATA study,[13] corroborate our results: The prevalence of AF is increasing with time in the northern region of Portugal. As far as following expectations, this might also be related to the improvement of the coding process of the problems in the clinical process. It was found that the majority of patients with AF in this study are elderly, as in the ATRIA study[4] (44.8% were aged 80 years or older vs 45% were aged 75 years or older).

The $CHA_2DS_2$-VASc scores found in our study are similar to those of the FATA study[13]: 0 points (2.6% vs 2.3%), 1–3 points (40.7% vs 40.9%), 4–5 points (43.3% vs 41.6%) and ≥6 (13.4% vs 14.3%). In our study, 44 326 (72.9%) patients were adequately anticoagulated, which is an improvement relative to the FAMA,[3] FATA,[13] SAFIRA[14] and REACH[20] studies (38%, 56.8%, 43.7% and 54%, respectively). These studies were published prior to the 2016 ESC guidelines.[19] The guidelines advise that NOAC should be considered for men with a $CHA_2DS_2$-VASc score ≥1 and women with a score ≥2, which might have contributed to the improvement in anticoagulation in patients with AF. It should be noted that, in our study, 79% of patients with AF who are not anticoagulated have a high thrombotic risk ($CHA_2DS_2$-VASc score ≥3). This information is of most importance because clinicians should be proactive in the prevention of stroke in patients with AF.

In addition to evidence of anticoagulation undertreatment in patients with AF, overtreatment is also found. In our study, 40.9% of patients with AF who do not have an indication for anticoagulation are taking anticoagulants. Similar overtreatment rates occur in the REACH study,[20] where 43% of patients with AF and not indicated for anticoagulation were receiving anticoagulants. The FAMA,[3] FATA[13] and SAFIRA[14] studies did not report overtreatment rates. A possible explanation for this overtreatment rate in our study might be a misuse of the $CHA_2DS_2$-VASc score. Some of these patients might also be prescribed an anticoagulation agent for other clinical reasons. Hess *et al*[21] identified a broad range of barriers to oral anticoagulation, including knowledge gaps about stroke risk and the relative risks and benefits of anticoagulant therapies; lack of awareness regarding the potential use of NOAC agents for VKA-unsuitable patients; lack of recognition of expanded eligibility for oral anticoagulation; lack of availability of reversal agents and the difficulty of anticoagulant effect monitoring for the NOACs; concerns with the bleeding risk of anticoagulant therapy, especially with the NOACs and particularly in the setting of dual antiplatelet therapy; suboptimal time in therapeutic range for VKA; and costs and insurance coverage.

Although there has been a positive evolution regarding the number of patients with anticoagulated AF, warfarin remains the most prescribed anticoagulant. As noted before, this is not concordant with the 2016 ESC guidelines.[19] When analysing the distribution of drugs by $CHA_2DS_2$-VASc score and gender, it should be noted that in men who are not indicated to be anticoagulated ($CHA_2DS_2$-VASc score of 0), the most prescribed drug is rivaroxaban, which might result from another therapeutic indication, such as deep venous thrombosis. On the other hand, acenocoumarol is more prescribed than dabigatran both in men with $CHA_2DS_2$-VASc scores of 7 and 8 and in women with a $CHA_2DS_2$-VASc score of 8. These results might be explained by the fact that clinicians prefer drugs with which they have more experience, such as VKA in patients with a very high thrombotic risk. However, this does not follow the most current guidelines.

The SAFIRA[14] study showed a very high prevalence of CV risk factors in patients with AF. In Portugal, SCORE is recommended for use in calculating CV risk in those between 40 and 65 years old.[22 23] However, we found no study that calculated CV risk by SCORE. The CV risk of patients aged 40–65 years was considered, with most patients having a very high CV risk. In fact, according to the 2016 ESC guidelines,[19] most patients with AF have many other comorbidities, so we consider this to be a reason for having a much larger number of very high risk than high-risk patients in our study.

The SAFIRA[14] study found that 24.7% of the patients were incorrectly medicated (inadequate dose, inadequate number of doses or creatinine clearance incompatible with the use of NOAC), but it did not discriminate patients by stages of CKD. The AF-React study showed that NOACs are being prescribed in stage 5 of CKD, which was contraindicated by the guidelines in force at the date of prescription (between 2016 and 2018).[18 24] In stage 4 of CKD, the 2016 ESC Guidelines[19] do not recommend the use of NOACs; the EHRA practical guide[24] recommends dose adjustment for apixaban, rivaroxaban and edoxaban and does not recommend the use of dabigatran. These data open the door to a more detailed analysis in the future about the prescriptions in stage 4.

## Limitations

The main limitation of this retrospective study is that the AF assessment is based on data coded in the clinical process in primary healthcare. There might be more patients with AF in the northern region of Portugal, but they are not coded at primary healthcare and, therefore, were not included in this study. Hence, strict coding by all family doctors is important. Another limitation might be the lack of ICPC-2 coding for problems such as mechanical prosthesis valve and rheumatic moderate or severe mitral stenosis.[19] There might be a small percentage of patients who are well medicated with VKA because they are in one of those situations.

The code K78, ICPC-2 included the diagnosis of AF and atrial flutter. Few of the 63 526 patients will be diagnosed

with atrial flutter, so this is a coding limitation of ICPC-2. However, this limitation does not invalidate the results of our study because, for patients with atrial flutter, anticoagulant therapy is recommended according to the same risk profile used for AF.[25]

The fact that it is not known when treatment started for patients already diagnosed with AF before 2016 may be a limitation for interpreting the data. Besides that, the database did not allow a detailed timing and sequencing of diagnosis, which might have implications for interpreting our findings.

### Implications for research and practice

The AF-React study made it possible to determine the prevalence of AF and to evaluate how patients with AF are treated in northern Portugal. Thus, the AF-React study brings extremely relevant conclusions to Portugal and follows real-world studies in patients with AF in Europe, presenting some data not yet studied: distribution of patients for $CHA_2DS_2$-VASc score value, gender and respective drugs, classification of CV risk and distribution of patients for CKD stages by respective drugs. For the future, it would be necessary for health authorities to allow continued access to these data sets, including mortality data, to draw conclusions (based on the real world) about the impact of the new treatments on patient-oriented outcomes.

#### Author affiliations

[1]Department of Community Medicine, Information and Health Decision Sciences (MEDCIDS), Faculty of Medicine of the University of Porto, Porto, Portugal
[2]Center for Health Technology and Services Research (CINTESIS), Faculty of Medicine of the University of Porto, Porto, Portugal
[3]Health Centre Grouping Santo Tirso/Trofa, Family Health Unit S. Tomé, Santo Tirso, Portugal
[4]Instituto Politécnico de Viana do Castelo, Viana do Castelo, Portugal
[5]Research and Planning Department, Regional Health Administration of Northern, Ministry of Health Portugal, Porto, Portugal

**Acknowledgements** A special thanks to all primary healthcare professionals who in their daily work in the Portuguese Health System fulfil the clinical records and allow researchers like us to obtain evidence that will help us to improve the health of the Portuguese.

**Contributors** SSP and CM contributed to the conception of the work. All authors contributed to the design of the study. HM contributed to the data acquisition process. SPP, ASCT and TSH contributed to the analysis and interpretation of the data. SSP drafted the manuscript, HM, ASCT, TSH and CM critically revised the manuscript. All authors gave final approval and agreed to be accountable for all aspects of the work, ensuring integrity and accuracy.

**Funding** This article was supported by National Funds through FCT - Fundação para a Ciência e a Tecnologia I.P., within CINTESIS, R&D Unit (reference UIDB/IC/4255/2020). The authors have received financial support of €4400 from the #H4A Primary Healthcare Research Network Scholars Programme research support.

**Competing interests** None declared.

**Patient and public involvement** Patients and/or the public were not involved in the design, or conduct, or reporting, or dissemination plans of this research.

**Patient consent for publication** Not required.

**Ethics approval** The study protocol was approved by the Health Ethics Committee of the Regional Health Administration of Northern Portugal.

**Provenance and peer review** Not commissioned; externally peer reviewed.

**Data availability statement** Data are available upon reasonable request. The data processing was carried out through the Department of Studies and Planning of the Regional Health Administration of Northern Portugal (Ministry of Health, Portugal). The data were extracted from the server through an anonymised data processing and editing platform and delivered securely (and in accordance with legal regulations and due approval) to the principal investigator. The data are property of Regional Health Administration of Northern Portugal (Ministry of Health, Portugal) and data analysis has been authorised for the purposes of the protocol only.

**ORCID iDs**
Susana Silva Pinto http://orcid.org/0000-0001-9764-6251
Andreia Teixeira http://orcid.org/0000-0003-1199-2220
Carlos Martins http://orcid.org/0000-0001-8561-5167

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
