## [Reviewer comments · BMJ Open]

ARTICLE DETAILS

TITLE (PROVISIONAL)	AF-React study – Atrial fibrillation management strategies in clinical practice: retrospective longitudinal study from real-world data in Northern Portugal
AUTHORS	Silva Pinto, Susana; Teixeira, Andreia; Henriques, Teresa; Monteiro, Hugo; Martins, Carlos

VERSION 1 – REVIEW

REVIEWER	Eitaro Kodani Nippon Medical School Tama-Nagayama Hospital, Tokyo, Japan.
REVIEW RETURNED	21-Jul-2020

GENERAL COMMENTS	This manuscript by Pinto et al. focused on the current status of anticoagulation therapy in patients with atrial fibrillation (AF) in Northern Portugal using data of the AF-React study. Authors demonstrated that the prevalence of AF, distribution of CHA2DS2-VASc score, and frequency of oral anticoagulants (OACs) in each CHA2DS2-VASc score. I agree it is important for physicians to know the information of real-world data. Therefore, the concept of this study is understandable and results seem reasonable. Although this manuscript seems written well, authors may want to consider several issues as below. Major comments; 1) Since the denominators of percentage were not defined properly, percentages described in this manuscript are not reliable. For example, authors described that “Of these, 44,326 (68.8%) are being treated with anticoagulants.” This means the denominator should be 60838 and $44326/60838=72.9\%$. In addition, authors described that “Albeit having an indication, there is a considerable proportion of patients (26.0%) that are not anticoagulated,” This means the denominator should be number of patients with CHA2DS2-VASc score of 1 or more for men and 2 or more for women. I cannot understand why it was 26.0%. Other percentages are also similar. 2) Although prevalence of AF was shown in Table 2, there was no description regarding the prevalence of AF in methods. Who were denominator subjects to calculate prevalence AF? Were they the general population or patients taking hospitals? How did authors calculate data shown in Table 2. These data are actually thought to be out of this study. 3) Although results of the present study seem important for physicians in your county Portugal, there was no novel or surprising information for those in other countries.
--

	Minor comments; 1) Abbreviations of CKD and RCV should be spelt out at the first time of use. 2) Table 1, total number of study population should be described. Mean and SD values of age are better to describe with the first decimal places. 3) In Table 3, recommendation criteria for anticoagulation therapy should be noted in the footnote. 4) Table 4, authors should consider description order of OACs. Acenocoumarol, one of vitamin K antagonist (VKA), should not be placed between non-VKA oral anticoagulants (NOACs). It seems better to show VKA group and NOAC group first, then show each OAC. Also, it seems better to show data of overall patients first, then show them each sex category. 5) In Table 5, data shown here are actually comorbidities or underlying diseases in patients with AF, not pathology. 6) Why didn't you show the frequency of OAC in each CKD stage?
--	---

REVIEWER	Tanja Mueller University of Strathclyde, United Kingdom
REVIEW RETURNED	19-Sep-2020

GENERAL COMMENTS	AF react study – review report Background Lines 100/101: 39% reduction in mortality – compared to what/when? Lines 106-100: NOACs have not consistently been shown to be superior to warfarin (according to meta-analyses and observational studies); they were mostly similar in terms of effectiveness, and sometimes slightly better with regards to bleeding events, probably depending on the patient and the specific drug used – suggest to add reference(s) here. Lines 111: “[...] the lower prevalence of these events with NOACs [...]” – compared to warfarin? Lines 113-15: This sentence is a bit unclear. Do you mean that the reduction in stroke (in recent years?) was probably due to changes in treatment guidelines, following the introduction of NOACs into clinical practice? Suggest to clarify this statement. Line 121: It might be helpful to highlight that all RCTs compared a NOAC to warfarin; there are no RCTs directly comparing NOACs to each other (or to placebo, for obvious reasons). Lines 127/128: That statement requires some elaboration – there have been quite a few studies done throughout Europe looking at various aspects of AF and oral anticoagulants; but perhaps not with a focus on some specific issues of interest? See e.g. Ibanez et al (2019), doi 10.1111/bcp.14071; Komen et al (2017), doi 10.1007/s00228-017-2289-0; Mueller et al (2019), doi: 10.1111/bcp.13814; Oimpieri et al (2020), doi: 10.1016/j.ijcha.2019.100465; Vinogradova et al (2018), doi: https://doi.org/10.1136/bmj.k2505 – these are just some of the papers that have been published over the last few years, focusing
---

	on various aspects of OAC/NOAC prescribing/use and/or effectiveness, and using large databases. Methods To clarify the study methods and potentially enable replication of the study in other settings, it would be helpful to add a few more details, including, e.g.: The database used: What is its original purpose; what data is included; and how/when/where is the data collected? Does it cover the entire population, or are specific patients missing for some reason? The cohort identification: The study includes “[...] adults with the code for AF [...] as an active problem between January 2016 and December 2018.”; does this only include patients who were diagnosed between 2016 and 2018, or also those that were diagnosed prior to this but had some contact with the health care system and therefore had the diagnosis recorded for some reason during this time period? What was considered to be a patient’s study inclusion (index/baseline) date? Perhaps a flowchart would be very helpful here. The patient characteristics: At what point in time was the CHA2DS2-VASc score calculated for each patient – at time of diagnosis/baseline date? And what was the look-back period to determine both the risks scores and other comorbidities? Were all the conditions listed in table 5 present at baseline, or could these have developed at any point in time after the patient appeared in the database (i.e. after an initial AF diagnosis/a first OAC prescription)? The risk assessments undertaken: How were the CHA2DS2-VASc score and the CVD risk SCORE calculated, and what data/variables were used for this? Also: what is the exact purpose of the SCORE – it is not specifically mentioned in the results, nor is it further discussed in the discussion section (except from noting its limitations in the current setting)? The exposure: For those patients with OAC prescriptions – what was the timing of AF and OAC; did all patients receive a first OAC prescription following a diagnosis? Were all patients new OAC users, or was the study population a mix of incident and prevalent OAC users (i.e. was it possible for patients to already be on OAC treatment prior to study start)? AF prevalence: A description of how this has been calculated is missing. Results Line 168: The age range is incomplete (range, to 107 years). Lines 181-184: It would be helpful to add a sentence here mentioning the drugs prescribed, similar to what is included the preceding paragraph. Lines 185-194: This paragraph is slightly confusing; would it be possible to simplify this, considering the detailed results are included in table 4? Lines 195-197: The accuracy of these results is dependent on whether patients were new users (initiating treatment during the study period); if patients were already on treatment, statements regarding therapy switches might be inaccurate. Some results are mentioned in the discussion, but are missing in the results section – e.g. the AF prevalence, or the distribution of CHA2DS2-VASc scores. I would suggest to move detailed findings to the results section for consistency.
--	--

	Table 1: What point in time do these variables refer to – the first time patients appear in the database with a recorded diagnosis of AF? I assume “professional situation” refers to employment status; what does “not applicable” mean in this context? And does employment status have any impact on access to health care, or medication? It might be useful to add BMI categories (underweight – obese); it appears that not very many patients have a health weight? Patients with a CHA2DS2-VASc score of 0 – could this include patients where the data required for its calculation was missing? Also: would it be possible to add columns stratifying patients by treatment status (untreated vs OAC, or maybe even VKA and NOAC separate)? Table 2: Could you please add numbers and denominators? Or, alternatively, mention those figures somewhere in the manuscript. Table 3: Are these recommendations solely based on CHA2DS2-VASc score? Table 4: It would be safe to remove the row in line 16 (males with a score of 9). Discussion It would be interesting to know when the individual NOACs were approved for use in Portugal (this might explain the huge differences in use between them). Also: are national/regional/local AF treatment guidelines in line with the ESC recommendations; and do guidelines specify which drug to use in any given circumstance? If local guidelines contain other treatment thresholds and/or recommendations with regards to specific drugs, deviations from ESC guidelines are to be expected. Another aspect related to over-/undertreated with OACs is the accuracy of AF diagnoses in the databases, as well as the presence/absence of potential alternative indications. While this has briefly been alluded to, it might be useful to expand on this, particularly the latter; for instance, the sequencing of AF diagnosis and OAC prescriptions; the duration of treatment; and/or the drug dosage prescribed might indicate treatment for reasons other than AF – including the already mentioned DVT, short-term therapy following hip- or knee replacement surgery, or ACS (rivaroxaban 2.5mg). Is there a way of potentially identifying alternative indications in the database?
--	--

VERSION 1 – AUTHOR RESPONSE

Reviewer: 1

Eitaro Kodani

Nippon Medical School Tama-Nagayama Hospital, Tokyo, Japan.

This manuscript by Pinto et al. focused on the current status of anticoagulation therapy in patients with atrial fibrillation (AF) in Northern Portugal using data of the AF-React study. Authors demonstrated that the prevalence of AF, distribution of CHA2DS2-VASc score, and frequency of oral anticoagulants (OACs) in each CHA2DS2-VASc score. I agree it is important for physicians to know the information of real-world data. Therefore, the concept of this study is understandable and results seem reasonable. Although this manuscript seems written well, authors may want to consider several issues as below.

Dear Reviewer,
Dr. Eitaro Kodani

We appreciate your overview of this manuscript. In fact, stroke remains the major cause of mortality related to vascular diseases in Portugal. Atrial fibrillation (AF) is currently responsible for about 15% of strokes, so we consider that it is important to carry out this study based on real-world data. In this way, physicians know the reality and we can show that we can still do better to reduce strokes resulting from AF.

Major comments:

1) Since the denominators of percentage were not defined properly, percentages described in this manuscript are not reliable. For example, authors described that “Of these, 44,326 (68.8%) are being treated with anticoagulants.” This means the denominator should be 60838 and $44326/60838=72.9\%$. In addition, authors described that “Albeit having an indication, there is a considerable proportion of patients (26.0%) that are not anticoagulated,” This means the denominator should be number of patients with CHA₂DS₂-VASc score of 1 or more for men and 2 or more for women. I cannot understand why it was 26.0%. Other percentages are also similar.

In fact, there was some confusion regarding some denominators for calculating percentages. We appreciate this comment. The percentages were all revised and changed accordingly in manuscript.

2) Although prevalence of AF was shown in Table 2, there was no description regarding the prevalence of AF in methods. Who were denominator subjects to calculate prevalence AF? Were they the general population or patients taking hospitals? How did authors calculate data shown in Table 2. These data are actually thought to be out of this study.

In fact, the method of calculating the prevalence of AF was not well described in the manuscript. Once again, we appreciate this precious comment. The prevalence of AF was calculated based on the age pyramid of the population enrolled in primary health care in northern Portugal in 2016, 2017 and 2018. Thus, the denominator for calculating the prevalence of AF was based on the age pyramid of this population. This information was added to the manuscript and is now well clarified (lines 153-154). In addition, Table 2 was modified: now containing denominators in all age groups to calculate the respective prevalence.

3) Although results of the present study seem important for physicians in your county Portugal, there was no novel or surprising information for those in other countries.

We understand this opinion and appreciate it. In fact, there are studies with real-world data on patients with AF in Europe, studying several variables and characteristics, related to various aspects NOAC prescribing/use and/or effectiveness. However, our study analyzes some data not found in other studies, namely, the distribution of patients for CHA₂DS₂-VASc score value, gender, and respective drugs, classification of Cardiovascular Risk and distribution of patients for Chronic Kidney Disease' stages by respective drugs. Thus, we included studies with real-world data in Europe with interesting results in the introduction (lines 131-141) and we added the following sentence in the discussion section (lines 311-312) and in the abstract “The AF-React study brings extremely relevant conclusions to Portugal and is following real-world studies in patients with AF in Europe, presenting some data not studied yet.”

Minor comments:

1) Abbreviations of CKD and RCV should be spelt out at the first time of use.

We fully agree that the abbreviations must be spelled out the first time they are used in order to facilitate the reading of the manuscript. We appreciate the detection of this lapse. It has already been corrected in the manuscript: CV risk (cardiovascular risk) – line 155. CKD (chronic kidney disease) – line 160.

2) Table 1, total number of study population should be described. Mean and SD values of age are better to describe with the first decimal places.

We agree with these suggestions and made the necessary changes that are signed in the manuscript.

3) In Table 3, recommendation criteria for anticoagulation therapy should be noted in the footnote. In fact, it is important to clarify in the table which criterias are recommended for anticoagulant therapy. We appreciate the suggestion. This information has already been added to the Table 3.

4) Table 4, authors should consider description order of OACs. Acenocoumarol, one of vitamin K antagonist (VKA), should not be placed between non-VKA oral anticoagulants (NOACs). It seems batter to show VKA group and NOAC group first, then show each OAC. We agreed that the presentation of Table 4 was a little confusing and we accepted the comment for improvement. Table 4 was reformulated according your suggestion, having been clarified. Also, it seems batter to show data of overall patients first, then show them each sex category. We are grateful for the suggestion, but taking into account that the CHA₂DS₂-VASc depends on sex (as women vary from 1 to 9, while men vary from 0 to 8), we consider that presenting the frequencies in general, in Table 4, could cause confusion, as the anticoagulation indication criterion is different depending on gender. In addition, Table 1, which represents the general characterization of the population, already presents general results for CHA₂DS₂-VASc categories, so we consider that it would be duplication of information.

5) In Table 5, data shown here are actually comorbidities or underlying diseases in patients with AF, not pathology.

We agree that Table 5 shows comorbidities or underlying diseases in patients with AF. Thus, they were changed and are marked in Table 5.

6) Why didn't you show the frequency of OAC in each CKD stage?

We are very grateful for this suggestion, which had not come to us before. In fact, it will be new data that greatly improve this manuscript. In this way, we proceeded to this analysis and included a new table with these data (Table 5), as well as added them in the results and in the discussion. All of these new data are highlighted in the manuscript.

Reviewer: 2

This is a very interesting and useful descriptive study, and the manuscript is (mostly) well written; some details with regards to methods are missing though, and the results/discussion is not entirely consistent with the introduction/methods section. I would recommend some revisions mainly to clarify the methods used (minor point: the references appear not to be the most up-to-date).

Please find my detailed comments in the file attached.

Dear reviewer,
Dr Tanja Mueller

First of all, we appreciate the recognition of the usefulness and interest of this manuscript and its extensive and detailed review that has brought a considerable improvement to our article. We are very grateful for that.

AF react study – review report

Background

- Lines 100/101: 39% reduction in mortality – compared to what/when?

We appreciate your request for clarification. In fact, it is not clear what the comparison is in the manuscript. Data on mortality reduction in 2015 refer to the comparison with data from 2011. This information was added in the manuscript (line 101).

- Lines 106-100: NOACs have not consistently been shown to be superior to warfarin (according to meta-analyses and observational studies); they were mostly similar in terms of effectiveness, and sometimes slightly better with regards to bleeding events, probably depending on the patient and the specific drug used – suggest to add reference(s) here.

We are grateful for the suggestion and agree to place the references of these studies in order to make the statement more robust. These references are marked in the text (line 111).

- Lines 111: “[...] the lower prevalence of these events with NOACs [...]” – compared to warfarin? Yes, the lower prevalence of these events with NOACs is compared to warfarin. This information has already been added to the manuscript and is highlighted (lines 112-113).

- Lines 113-15: This sentence is a bit unclear. Do you mean that the reduction in stroke (in recent years?) was probably due to changes in treatment guidelines, following the introduction of NOACs into clinical practice? Suggest to clarify this statement.

We agree that, in fact, the sentence is confused. The sentence has been reformulated and we believe that it is now more clarified (lines 115-116). We thank the comment.

- Line 121: It might be helpful to highlight that all RCTs compared a NOAC to warfarin; there are no RCTs directly comparing NOACs to each other (or to placebo, for obvious reasons). In fact, the information that there are no RCTs to compare NOAC with each other or with placebo has not been clarified. We appreciate the suggestion. This information has been added and is noted in the manuscript (lines 122-123).

- Lines 127/128: That statement requires some elaboration – there have been quite a few studies done throughout Europe looking at various aspects of AF and oral anticoagulants; but perhaps not with a focus on some specific issues of interest? See e.g. Ibanez et al (2019), doi 10.1111/bcp.14071; Komen et al (2017), doi 10.1007/s00228-017-2289-0; Mueller et al (2019), doi: 10.1111/bcp.13814; Olimpieri et al (2020), doi: 10.1016/j.ijcha.2019.100465; Vinogradova et al (2018), doi: <https://doi.org/10.1136/bmj.k2505> – these are just some of the papers that have been published over the last few years, focusing on various aspects of OAC/NOAC prescribing/use and/or effectiveness, and using large databases.

We are grateful for the suggestion of more recent studies to be included in the literature review. A new literature review was carried out and European studies related to our study were included. The new paragraph is highlighted (lines 131-141).

Methods

To clarify the study methods and potentially enable replication of the study in other settings, it would be helpful to add a few more details, including, e.g.:

- The database used: What is its original purpose; what data is included; and how/when/where is the data collected? Does it cover the entire population, or are specific patients missing for some reason? We understand the issue. In fact, the matter was unclear. The first paragraph of the methods section has been reformulated so that the question is made explicit (146-152). We thank the comment.

- The cohort identification: The study includes “[...] adults with the code for AF [...] as an active problem between January 2016 and December 2018.”; does this only include patients who were

diagnosed between 2016 and 2018, or also those that were diagnosed prior to this but had some contact with the health care system and therefore had the diagnosis recorded for some reason during this time period? What was considered to be a patient's study inclusion (index/baseline) date? Perhaps a flowchart would be very helpful here.

The authors appreciate and thank the comment. The database included clinical information from 63526 patients with AF. "Active problem" means that the patient has the disease at the time the data collected from the clinical process. For each patient, the clinical information collected was the last available between January 2016 and December 2018, so it was for these data that the quality was guaranteed.

- The patient characteristics: At what point in time was the CHA₂DS₂-VASc score calculated for each patient – at time of diagnosis/baseline date? And what was the look-back period to determine both the risks scores and other comorbidities? Were all the conditions listed in table 5 present at baseline, or could these have developed at any point in time after the patient appeared in the database (i.e. after an initial AF diagnosis/a first OAC prescription)?

Once again, we appreciate the point of view. For each patient, the clinical information collected was the last available between January 2016 and December 2018, so it was for these data that the quality was guaranteed. Thus, this methodology happened for the CHA₂DS₂-VASc score and for all other variables. All the conditions listed in Table 5 (now, Table 6) were the ones that were "active problem", after the diagnosis of AF. "Active problem" means that the patient has the disease at the time the data is collected from the clinical process.

- The risk assessments undertaken: How were the CHA₂DS₂-VASc score and the CVD risk SCORE calculated, and what data/variables were used for this? Also: what is the exact purpose of the SCORE – it is not specifically mentioned in the results, nor is it further discussed in the discussion section (except from noting its limitations in the current setting)?

We thank the request for clarification. The CHA₂DS₂-VASc score was calculated according to the 2016 ESC guidelines for AF, as follows by ICPC-2 coding (ICPC-2 is primary health care coding). So CHA₂DS₂-VASc was calculated according to ICPC-2 codes, placed here in parentheses, that were being coded in clinical process: C-Congestive heart failure (K77): 1 point; H-Hypertension (K86 or K87): 1 point; A2-Age > 75 years or older: 2 points; D-Diabetes mellitus (T89 or T90): 1 point; S2-Stroke (K89, K90 or K91): 2 points; V-Vascular disease (K75 or K92): 1 point; A-Age 65-74 years: 1point; Sc-Sex category (female): 1 point. This clarification has been added and is noted in the manuscript (lines 169-172).

Cardiovascular risk was defined in two ways: on the one hand, by extracting the percentage of the SCORE from the clinical process and classified into four stages as indicated in the manuscript (lines 155-159), and on the other hand, through variables that alone define a high and very high cardiovascular risk, as indicated in the manuscript (lines 159-163).

This suggestion has already been considered and we have already added a better description to the methodology (lines 155-163). In the results section, was added information on these data (lines 219-221), which had already been the subject of discussion (lines 288-295).

- The exposure: For those patients with OAC prescriptions – what was the timing of AF and OAC; did all patients receive a first OAC prescription following a diagnosis? Were all patients new OAC users, or was the study population a mix of incident and prevalent OAC users (i.e. was it possible for patients to already be on OAC treatment prior to study start)?

We consider these questions to be very interesting and pertinent, so we thank you for your attention. Our database includes only patients between 2016 and 2018. There are many patients who had a diagnosis of AF prior to 2016, however we were unable to obtain these data prior to 2016, so we were unable to answer these questions.

- AF prevalence: A description of how this has been calculated is missing.

In fact, the method of calculating the prevalence of AF was not well described in the manuscript. The prevalence of AF was calculated based on the age pyramid of the population enrolled in primary health care in northern Portugal in 2016, 2017 and 2018. Thus, the denominator for calculating the prevalence of AF was based on the age pyramid of this population. This information was added to the manuscript and is now well clarified (lines 153-154). In addition, Table 2 was modified: now containing denominators in all age groups to calculate the respective prevalence.

Results

- Line 168: The age range is incomplete (range, to 107 years).

Thanks for this correction. The change has been made and is noted in the manuscript (line 192).

- Lines 181-184: It would be helpful to add a sentence here mentioning the drugs prescribed, similar to what is included in the preceding paragraph.

We agree that it can be useful, so the description of the drugs prescribed by the two classes: VKAs and NOACs has been added and is marked in the manuscript (lines 202-203).

- Lines 185-194: This paragraph is slightly confusing; would it be possible to simplify this, considering the detailed results are included in table 4?

We appreciate the suggestion and agree that this paragraph is a little confusing. The paragraph has been reworded to keep it simple (lines 212-218).

- Lines 195-197: The accuracy of these results is dependent on whether patients were new users (initiating treatment during the study period); if patients were already on treatment, statements regarding therapy switches might be inaccurate.

We understand this point of view. The database of this study includes only patients between 2016 and 2018, we were unable to obtain these data prior to 2016, so we were unable to know if patients were already on treatment.

- Some results are mentioned in the discussion, but are missing in the results section – e.g. the AF prevalence, or the distribution of CHA₂DS₂-VASc scores. I would suggest to move detailed findings to the results section for consistency.

We are grateful for the comment, having completed the results section: the AF prevalence – lines 193-196; the distribution of CHA₂DS₂-VASc – lines 206-207.

- Table 1: What point in time do these variables refer to – the first time patients appear in the database with a recorded diagnosis of AF? I assume “professional situation” refers to employment status; what does “not applicable” mean in this context? And does employment status have any impact on access to health care, or medication? (4) It might be useful to add BMI categories (underweight – obese); it appears that not very many patients have a health weight? Patients with a CHA₂DS₂-VASc score of 0 – could this include patients where the data required for its calculation was missing? Also: would it be possible to add columns stratifying patients by treatment status (untreated vs OAC, or maybe even VKA and NOAC separate)?

We appreciate the questions and suggestions. For each patient, the clinical information collected was the last available between January 2016 and December 2018, so it was for these data that the quality was guaranteed. In fact, we considered the suggestion and added the category of the professional situation “not applicable” to the category “unknown”. The changes were made to the manuscript (Table 1). We agree with the reviewer, in fact the employment status have any impact on access to health care or medication, so it was studied. We agree with the suggestion to make known the BMI categories (underweight – obese). This was included in Table 1. In fact, in this manuscript we are dependent on the coding of doctors in clinical processes. However, as the pathologies included in the CHA₂DS₂-VASc score are important pathologies, which family doctors should pay special attention to, we consider that coding errors are rare. We understand the

suggestion of stratification patients by treatment status, however, throughout the study period (between January 2016 and December 2018), the same patient may belong to more than one group, as the drug may not have been maintained. Thus, we consider that it could generate some confusion in the presentation of these data.

- Table 2: Could you please add numbers and denominators? Or, alternatively, mention those figures somewhere in the manuscript.

We are very grateful for the suggestion. Table 2 was reformulated, according to your suggestion.

- Table 3: Are these recommendations solely based on CHA2DS2-VASc score?

In fact, it is important to clarify in the table which criteria are recommended for anticoagulant therapy. We appreciate the suggestion. This information has already been added to the Table 3.

- Table 4: It would be safe to remove the row in line 16 (males with a score of 9).

We agree with this suggestion. We remove the row in line 16 (males with a score of 9) in Table 4.

Discussion

It would be interesting to know when the individual NOACs were approved for use in Portugal (this might explain the huge differences in use between them). Also: are national/regional/local AF treatment guidelines in line with the ESC recommendations; and do guidelines specify which drug to use in any given circumstance? If local guidelines contain other treatment thresholds and/or recommendations with regards to specific drugs, deviations from ESC guidelines are to be expected. We appreciate the suggestions that are extremely useful. The apixaban, dabigatran and rivaroxaban were approved for use and reimbursed by the national health service in 2010 and edoxaban in 2017, in Portugal. Therefore, this difference may explain the lower rate of use of edoxaban. However, it does not explain the differences in use between dabigatran, apixaban and rivaroxaban, since they were reimbursed at the same time. Regarding local guidelines, in Portugal there are no guidelines for the treatment of AF, so doctors are guided by ESC recommendations.

Another aspect related to over-/undertreated with OACs is the accuracy of AF diagnoses in the databases, as well as the presence/absence of potential alternative indications. While this has briefly been alluded to, it might be useful to expand on this, particularly the latter; for instance, the sequencing of AF diagnosis and OAC prescriptions; the duration of treatment; and/or the drug dosage prescribed might indicate treatment for reasons other than AF – including the already mentioned DVT, short-term therapy following hip- or knee replacement surgery, or ACS (rivaroxaban 2.5mg). Is there a way of potentially identifying alternative indications in the database?

We are very appreciative of this comment and suggestion. In fact, it will be very interesting to carry out this analysis, it makes perfect sense in our line of research. However, at the moment, we do not have this data to do so. We will certainly follow the suggestion and try in the near future to access this data and carry out the suggested analysis. We were really grateful for the suggestion.

VERSION 2 – REVIEW

REVIEWER	Eitaro Kodani Nippon Medical School Tama Nagayama Hospital, Tokyo, Japan.
REVIEW RETURNED	28-Oct-2020

GENERAL COMMENTS	This revised manuscript by Pinto et al. focused on the current status of anticoagulation therapy in patients with atrial fibrillation (AF) in Northern Portugal using data of the AF-React study. Authors demonstrated that the prevalence of AF, distribution of
---

	CHA2DS2-VASc score, and frequency of oral anticoagulants (OACs) in each CHA2DS2-VASc score. Although authors revised the manuscript according to the reviewers' comments, there are several concerns to be resolved. Major comments; 1) Although authors showed the number of denominators for calculating the prevalence of AF, it remains unclear in detail. Descriptions about it are different between main text and footnote in Table 2. Were they the general population or patients taking hospitals? When the prevalence of AF is evaluated, what population and characteristics of denominator subjects are extremely important. If they are unclear, the prevalence of AF presented here cannot be evaluated or compared with that in other reports. Another reviewer also pointed out the same things. Thus, these issues should be clarified essentially, if authors want to show the prevalence of AF. Otherwise, it is better to focus this study on only characteristics of 63,526 patients with AF. Minor comments; 1) In Results, age should be 75.6±10.6 as that in Table 1. 2) In Table 1, definition of BMI category, stage of CKD, and classification of cardiovascular risk should be clarified and described. 3) In Table 1, categorization of CHA2DS2-VASc score such as 0, 1-3, 4,5, and ≥6 is uncommon and seems inappropriate. 4) Table 4, it seems better to show NOAC group, VKA group, and no OAC group first, then show each OAC.
--	---

REVIEWER	Tanja Mueller University of Strathclyde, UK
REVIEW RETURNED	09-Nov-2020

GENERAL COMMENTS	This descriptive study is potentially very relevant to clinical practice since it offers important insights into both AF prevalence and oral anticoagulant prescribing practice in Portugal. While the study design seems appropriate for its purpose, there are a few aspects that might be worth addressing in more detail though in order to support reproducibility of the method and interpretability of results. 1. the database: it is still a bit unclear what kind of data is included in the database used for the study, and whether this covers the entire population of the study region (e.g. variables, time frames covered) 2. the cohort: just for clarification - the study cohort comprised of all patients that had a relevant diagnostic code recorded within the database between 2016 and 2018 - and that potentially includes both prevalent and incident patients? 3. the co-variables: what data was used to calculate the risk scores (i.e. what kind of variables), and what time frame has been used? (i.e. referring to which point in time; there is a reference to the last record available, does this relate to the risk score calculations?) It seems data was available for patients to calculate the CHADS-VASc score, but not the CV risk score - why was that? 4. the exposure: what kind of data was used to identify whether patients have been prescribed oral anticoagulants (prescription
--

	records, claims data)? How has the appropriateness of the prescribing been established (i.e. what was the sequencing of AF diagnosis, data used to calculate risk scores, and first prescription)? In line 197 e.g., it says that 95.8% of patients had an indication for an OAC prescription; is this based solely on 2016-2018 data? And does this potentially include patients where prescriptions were issued prior to AF having been recorded in the database? 5. the study limitations: there is no discussion of the study limitations within the discussion section (e.g. regarding assumptions made, or data availability/quality; and the potential implications on findings might be - such as alternative indications for treatment, underestimation of co-morbidities based on short time frames, etc.) Minor comments:  - abstract: suggest to also include the potential undertreatment of patients in the results section; it seems a bit odd to introduce new findings in the conclusion - lines 117-121: it would make it easier for the reader if the years the studies (FATA/SAFITA) were conducted would be mentioned in the text - line 181: what are these specific filters mentioned - what was their purpose? suggest to delete if not relevant - line 200: I was wondering about the relatively high CHADS-VASc scores of the patients who were eligible for OACs but did not receive a prescription. Any suggestions why this might be? Could patients have received prescriptions that did not appear in the database (e.g. privately)? Or perhaps the risk factors were recorded just before the end of the study period, and the initial prescription was just not covered by the available data? - lines 219-221: how come there are so few patients classified as high risk? Also: the denominator here is difficult to figure out since the age classification used differs from what is presented in the baseline table - lines 227-230: were there no patients who switched from a NOAC to a VKA? And what was the proportion of patients with only a single recorded prescription for any OAC?
--	---

VERSION 2 – AUTHOR RESPONSE

Reviewer: 1

Reviewer Name: Eitaro Kodani

Institution and Country: Nippon Medical School Tama Nagayama Hospital, Tokyo, Japan.

Please state any competing interests or state 'None declared': None.

Comments to the Author:

This revised manuscript by Pinto et al. focused on the current status of anticoagulation therapy in patients with atrial fibrillation (AF) in Northern Portugal using data of the AF-React study. Authors demonstrated that the prevalence of AF, distribution of CHA2DS2-VASc score, and frequency of oral

anticoagulants (OACs) in each CHA2DS2-VASc score. Although authors revised the manuscript according to the reviewers' comments, there are several concerns to be resolved.

Dear Reviewer,

Dr. Eitaro Kodani

We appreciate your new overview of this manuscript. We believe that your new comments were very helpful in improving our manuscript.

Major comments:

1) Although authors showed the number of denominators for calculating the prevalence of AF, it remains unclear in detail. Descriptions about it are different between main text and footnote in Table 2. Were they the general population or patients taking hospitals? When the prevalence of AF is evaluated, what population and characteristics of denominator subjects are extremely important. If they are unclear, the prevalence of AF presented here cannot be evaluated or compared with that in other reports.

Another reviewer also pointed out the same things. Thus, these issues should be clarified essentially, if authors want to show the prevalence of AF. Otherwise, it is better to focus this study on only characteristics of 63,526 patients with AF.

We appreciate this precious comment. The descriptions of the denominators to calculate the prevalence were corrected in footnote of Table 2:

Lines 459-462: "#Population enrolled in primary health care in northern Portugal based on the age pyramid of Regional Health Administration of Northern Portugal in 2016. \$Population enrolled in primary health care in northern Portugal based on the age pyramid of Regional Health Administration of Northern Portugal in 2017. £Population enrolled in primary health care in northern Portugal based on the age pyramid of Regional Health Administration of Northern Portugal in 2018.

To clarify the denominator, we changed the first paragraph of the results section by adding the following text:

Lines 201-205: "Considering the 2,077,599 adults over 40 years enrolled in 2018 in the Regional Health Administration of Northern Portugal, the prevalence of AF was 3.044% (Table 2). In 2017, among the 1,943,813 adults over 40 years enrolled in the Regional Health Administration of Northern Portugal, the prevalence of AF was 2.787%. In 2016, among the 2,024,390 adults over 40 years enrolled in the Regional Health Administration of Northern Portugal, the prevalence of AF was 2.295%."

Minor comments:

1) In Results, age should be 75.6 ± 10.6 as that in Table 1.

We appreciate this correction. The age has been corrected in results section:

Lines 199-200: "The average age was 76.5 ± 10.6 years (range from 18 to 107 years)"

2) In Table 1, definition of BMI category, stage of CKD, and classification of cardiovascular risk should be clarified and described.

We thank this suggestion. We added the following text in the footnote of Table 1:

Lines 446-453: “# The body mass index is defined under six categories: <18,5 kg/m² – underweight; [18-25[kg/m² – normal; [25-30[kg/m² – overweight; [30-35[kg/m² – obese class I; [35-40[kg/m² – obese class II; ≥ 40 kg/m² kg/m² – obese class III.

* The stages of CKD were defined based on the glomerular filtration rate (GFR) calculated by the equation of Cockcroft-Gault: stage 1, GFR ≥ 90 ml/min; stage 2, GFR]90-60] ml/min; stage 3, GFR]60-30] ml/min, stage 4, GFR [30-15] ml/min; and stage 5, GFR < 15ml/min.

¥ The cardiovascular risk was defined under four stages that were based on SCORE (Systematic Coronary Risk Evaluation), as low, moderate, high, and very high, if less than 1%, between 1% and 5%, between 5% and 10%, and greater than or equal to 10%.”

3) In Table 1, categorization of CHA₂DS₂-VASc score such as 0, 1-3, 4,5, and ≥6 is uncommon and seems inappropriate.

We agree with this comment. In fact, the description by all possible scores (0, 1, 2, 3, 4, 5, 6, 7, 8 and 9) is more correct. This was modified in Table 1, variable CHA₂DS₂-VASc score.

CHA ₂ DS ₂ -VASc score, n(%)	
0	1658 (2.6%)
1	3967 (6.2%)
2	8065 (12.7%)
3	13811 (21.7%)
4	16152 (25.4%)
5	11388 (17.9%)
6	5520 (8.7%)
7	2196 (3.5%)
8	685 (1.1%)
9	84 (0.1%)

4) Table 4, it seems batter to show NOAC group, VKA group, and no OAC group first, then show each OAC.

We appreciate the comment, but we are not sure if we understand it well. In Table 4, we added the information by groups no OAC, NOAC and VKA. We replaced Table 4 by the following table:

Table 4. Distribution of patients for CHA₂DS₂-VASc score value, gender, and respective drugs.

Drugs, n (%)									
CHA ₂ DS ₂ - VASc score	No OA C*	NOAC# group					VKA\$ group		
	Total	Total	rivaroxab an	apixab an	dabigatr an	edoxab an	Total	warfar in	acenocoum arol
Male									
0	990 (59.7)	500 (30.2)	248 (15.0)	134 (8.1)	88 (5.3)	30 (1.8)	168 (10.1)	127 (7.7)	41 (2.5)
1	1069 (36.4)	1313 (44.7)	623 (21.2)	344 (11.7)	276 (9.4)	70 (2.4)	555 (18.9)	426 (14.5)	129 (4.4)
2	1646 (28.6)	2648 (46.0)	1125 (19.5)	801 (13.9)	590 (10.2)	132 (2.3)	1468 (25.5)	1099 (19.1)	369 (6.4)
3	2132 (25.3)	3742 (44.5)	1531 (18.2)	1210 (14.4)	823 (9.8)	178 (2.1)	2537 (30.2)	1955 (23.2)	582 (6.9)
4	1549 (24.8)	2572 (41.2)	1037 (16.6)	877 (14.0)	547 (8.8)	111 (1.8)	2127 (34.0)	1636 (26.2)	491 (7.9)
5	798 (25.0)	1301 (40.7)	471 (14.7)	486 (15.2)	286 (9.0)	58 (1.8)	1095 (34.3)	813 (25.5)	282 (8.8)
6	377 (27.1)	502 (36.1)	183 (13.2)	187 (13.4)	102 (7.3)	30 (2.2)	512 (36.8)	371 (26.7)	141 (10.1)
7	96	164	61 (14.7)	58 (13.9)	38 (9.1)	7 (1.7)	156	116 (27.9)	40 (9.6)

	(23 .1)	(39.4)					(37.5)		
8	19 (26 .4)	25 (34.7)	12 (16.7)	8 (11.1)	2 (2.8)	3 (4.2)	28 (38.9)	20 (27.8)	8 (11.1)
Subtotal	86 76 (28 .8)	1276 7 (42.4)	5291 (17.6)	4105 (13.6)	2752 (9.1)	619 (2.1)	8646 (28.7)	6563 (21.8)	2083 (6.9)
Female									
1	59 8 (58 .1)	272 (26.4)	118 (11.5)	88 (8.5)	49 (4.8)	17 (1.7)	160 (15.5)	125 (12.1)	35 (3.4)
2	75 7 (32 .9)	1004 (43.6)	418 (18.2)	345 (15.0)	189 (8.2)	52 (2.3)	542 (23.5)	432 (18.8)	110 (4.8)
3	15 44 (28 .6)	2399 (44.4)	951 (17.6)	781 (14.5)	515 (9.5)	152 (2.8)	1457 (27.0)	1130 (20.9)	327 (6.1)
4	25 55 (25 .8)	4460 (45.0)	1708 (17.2)	1565 (15.8)	979 (9.9)	208 (2.1)	2889 (29.2)	2195 (22.2)	694 (7.0)
5	21 42 (26 .1)	3369 (41.3)	1287 (15.7)	1281 (15.6)	694 (8.5)	152 (1.9)	2638 (32.4)	2035 (24.8)	603 (7.4)
6	11 17 (27 .1)	1666 (40.3)	627 (15.2)	623 (15.1)	330 (8.0)	86 (2.1)	1346 (32.6)	1024 (24.8)	322 (7.8)
7	52 5 (40.6)	722 (40.6)	232 (13.0)	312 (17.5)	150 (8.4)	28 (1.6)	533 (29.9)	398 (22.4)	135 (7.6)

	(29 .5)								
8	16 1 (26 .3)	257 (41.9)	85 (13.9)	121 (19.7)	43 (7.0)	8 (1.3)	195 (31.8)	148 (24.1)	47 (7.7)
9	25 (29 .8)	33 (39.3)	11 (13.1)	14 (16.7)	6 (7.1)	2 (2.4)	26 (31.0)	23 (27.4)	3 (3.6)
Subtotal	94 24 (28 .2)	1422 7 (42.5)	5437 (16.3)	5130 (15.3)	2955 (8.8)	705 (2.1)	9786 (29.3)	7510 (22.5)	2276 (6.8)
Total	18 10 0 (28 .5)	2699 4 (42.5)	10728 (16.9)	9235 (14.5)	5707 (9.0)	1324 (2.1)	1843 2 (29.0)	14073 (22.2)	4359 (6.9)

*OAC: oral anticoagulation

NOAC: non-vitamin K antagonist oral anticoagulants

\$ VKA: vitamin K antagonists

Reviewer: 2

Reviewer Name: Tanja Mueller

Institution and Country: University of Strathclyde, UK

Please state any competing interests or state 'None declared': None declared.

Comments to the Author:

This descriptive study is potentially very relevant to clinical practice since it offers important insights into both AF prevalence and oral anticoagulant prescribing practice in Portugal. While the study design seems appropriate for its purpose, there are a few aspects that might be worth addressing in more detail though in order to support reproducibility of the method and interpretability of results.

1. the database: it is still a bit unclear what kind of data is included in the database used for the study, and whether this covers the entire population of the study region (e.g. variables, time frames covered).

We thank for this comment. We added the following text at the beginning of the Methods section:

Lines 139-157: "The Portuguese National Health Service is a universal health coverage system and is administratively divided into five regions: North, Centre, Lisbon and Tagus Valley, Alentejo, and Algarve. The entirety of the mainland Portuguese population is enrolled in one of these administrative health regions. Each regional health administration is responsible for providing primary and secondary healthcare to the population living in its geographic area. This retrospective longitudinal study was conducted using data from the Regional Health Administration database of Northern Portugal.

The Department of Studies and Planning of the Regional Health Administration of Northern Portugal built a database containing the following data for the years 2016, 2017, and 2018: population enrolled in the Regional Health Administration of Northern Portugal, number of patients diagnosed with AF, and, for each AF patient, age, gender, professional situation, anticoagulant prescription history, other ICPC-2 coded health problems, and last record available of body mass index, cardiovascular risk score, and glomerular filtration rate. In fact, this is one-off data (once a year) for all variables except the anticoagulants prescription history, for which we had longitudinal information with the date of each prescription.

To identify AF patients, the International Classification of Primary Care (ICPC-2) code for AF, K78, was used.⁽¹⁹⁾ Therefore, we included all adults (age ≥ 18 years) with the code K78 between January and December of the years 2016, 2017, and 2018. For each year, either previous AF diagnosed patients or new AF diagnosed patients were included, so the database comprises both incident and prevalent patients."

2. the cohort: just for clarification - the study cohort comprised of all patients that had a relevant diagnostic code recorded within the database between 2016 and 2018 - and that potentially includes both prevalent and incident patients?

We thank this comment and to clarify this aspect, we added the following text:

Lines 155-157: "For each year, either previous AF diagnosed patients or new AF diagnosed patients were included, so the database comprises both incident and prevalent patients."

3. the co-variables: what data was used to calculate the risk scores (i.e. what kind of variables), and what time frame has been used? (i.e. referring to which point in time; there is a reference to the last record available, does this relate to the risk score calculations?) It seems data was available for patients to calculate the CHADS-VASc score, but not the CV risk score - why was that?

We thank this comment.

As it was described in the Methods section, the cardiovascular risk score was integrated in the database that we received from the North Health Region Administration. We hope that this is more clear in the following text that we added to the Methods section:

Lines 145-152: "The Department of Studies and Planning of the Regional Health Administration of Northern Portugal built a database containing the following data for the years 2016, 2017, and 2018: population enrolled in the Regional Health Administration of Northern Portugal, number of patients diagnosed with AF, and, for each AF patient, age, gender, professional situation, anticoagulant

prescription history, other ICPC-2 coded health problems, and last record available of body mass index, cardiovascular risk score, and glomerular filtration rate. In fact, this is one-off data (once a year) for all variables except the anticoagulants prescription history, for which we had longitudinal information with the date of each prescription.”

And:

Lines 160-163: “The CV risk (cardiovascular risk) was defined in four stages based on SCORE (Systematic Coronary Risk Evaluation), as low, moderate, high, and very high, if less than 1%, between 1% and 5%, between 5% and 10%, and greater than or equal to 10%, respectively, for patients between 40 and 65 years old.”

4. the exposure: what kind of data was used to identify whether patients have been prescribed oral anticoagulants (prescription records, claims data)? How has the appropriateness of the prescribing been established (i.e. what was the sequencing of AF diagnosis, data used to calculate risk scores, and first prescription)? In line 197 e.g., it says that 95.8% of patients had an indication for an OAC prescription; is this based solely on 2016-2018 data? And does this potentially include patients where prescriptions were issued prior to AF having been recorded in the database?

Once again, we appreciate your comment.

As we mentioned before, in the database we had information about the anticoagulants prescribed for each patient. These were the data we used to identify whether patients have been prescribed oral anticoagulants. About the appropriateness of the prescribing, in the Methods section we rewrote the following information:

Lines 176-178: “According to the 2016 ESC guidelines,(18) we considered a CHA₂DS₂-VASc score \geq 1 point in a male and a CHA₂DS₂-VASc score \geq 2 points in a female to be an indication of anticoagulation.”.

About the questions “In line 197 e.g., it says that 95.8% of patients had an indication for an OAC prescription; is this based solely on 2016-2018 data? And does this potentially include patients where prescriptions were issued prior to AF having been recorded in the database?”, yes that statement was based solely on 2016-2018 data, and no, we did not include prescriptions prior AF record.

5. the study limitations: there is no discussion of the study limitations within the discussion section (e.g. regarding assumptions made, or data availability/quality; and the potential implications on findings might be - such as alternative indications for treatment, underestimation of co-morbidities based on short time frames, etc.)

In fact, the limitations only appeared in the "Strengths and limitations of this study" section after the abstract. We recognize that it will be an improvement to put these limitations in the Discussion section. We added in Discussion section the following text:

Lines 314-325: “Limitations

The main limitation of this retrospective study is that the AF assessment is based on data coded in the clinical process in Primary Health Care. There might be more patients with AF in the northern region of Portugal, but they are not coded at Primary Health Care and, therefore, were not included in this study. Hence, strict coding by all family doctors is important. Another limitation might be the lack of ICPC-2 coding for problems such as mechanical prosthesis valve and rheumatic moderate or

severe mitral stenosis(18). There might be a small percentage of patients who are well medicated with vitamin K antagonists because they are in one of those situations.

The code K78, ICPC-2 included the diagnosis of AF and atrial flutter. Few of the 63,526 patients will be diagnosed with atrial flutter, so this is a coding limitation of ICPC-2. However, this limitation does not invalidate the results of our study because, for patients with atrial flutter, anticoagulant therapy is recommended according to the same risk profile used for AF.(25)”

Minor comments:

1. abstract: suggest to also include the potential undertreatment of patients in the results section; it seems a bit odd to introduce new findings in the conclusion.

We appreciate your suggestion. We thought that the potential undertreatment of patients was already included in the results of the abstract, but overtreatment was not included, so we appreciate your suggestion and have included this in the results section of the abstract. In fact, it seems a bit odd to introduce new findings in the conclusion. We added the following text:

Lines 64-65: “On the other hand, 2688 patients of the total (4.2%) had no indication to receive anticoagulation therapy. Of these 2688 patients, 1100 (40.9%) were receiving anticoagulants.”

2. lines 117-121: it would make it easier for the reader if the years the studies (FATA/SAFITA) were conducted would be mentioned in the text.

We agree with this suggestion and the year of the FATA and SAFIRA studies were added in the text:

Lines: 112-115: “In a cross-sectional study of patients with AF done in eight Family Health Units of Northern Portugal in 2015 (the FATA study),(13) only 56.8% of patients with AF were prescribed adequate oral anticoagulation according to the recommendations of the European Society of Cardiology (ESC) Guidelines in 2010. In the SAFIRA study (2018),(14) 56.3% of patients with AF were not anticoagulated.”

3. line 181: what are these specific filters mentioned - what was their purpose? suggest to delete if not relevant.

Thank you for raising this issue. In fact, filters are part of the data extraction process of the regional health administration in northern Portugal. Basically, it is the computer coding that allows to extract the variables that the researchers included in the study. We replaced the sentence by the following sentence:

Lines 186-187: “Data were obtained without aggregation at the individual level for general characterisation analysis.”

4. line 200: I was wondering about the relatively high CHADS-VASc scores of the patients who were eligible for OACs but did not receive a prescription. Any suggestions why this might be? Could patients have received prescriptions that did not appear in the database (e.g. privately)? Or perhaps the risk factors were recorded just before the end of the study period, and the initial prescription was just not covered by the available data?

Thank you for raising this question, which has enriched the manuscript.

In fact, we were also surprised by this result. We think that this work is very important to alert about this reality. In this context, we add the following sentence in the discussion section:

Lines 281-288: "Hess, P.L. et al. (2014) identified a broad range of barriers to oral anticoagulation, including knowledge gaps about stroke risk and the relative risks and benefits of anticoagulant therapies; lack of awareness regarding the potential use of NOAC agents for VKA-unsuitable patients; lack of recognition of expanded eligibility for oral anticoagulation; lack of availability of reversal agents and the difficulty of anticoagulant effect monitoring for the NOACs; concerns with the bleeding risk of anticoagulant therapy, especially with the NOACs and particularly in the setting of dual antiplatelet therapy; suboptimal time in therapeutic range for VKA; and costs and insurance coverage.(21)"

PS: We do not believe that the private prescriptions and/or prescriptions not covered by the available data had a major role.

5. lines 219-221: how come there are so few patients classified as high risk? Also: the denominator here is difficult to figure out since the age classification used differs from what is presented in the baseline table.

We appreciate your comment. In order to clarify the denominators, we updated Table 1:

Cardiovascular risk for patients with age between 40-65 years (n =8985), n(%)	
Value available	6531 (72.7%)
Missing record	2454 (27.3%)
Classification of Cardiovascular Risk* (n =6531), n(%)	
Low	650 (10.0%)
Moderate	2631 (40.3%)
High	194 (3.0%)
Very high	3056 (46.7%)

About the few number of patients classified as high risk, given most AF patients have other comorbidities and considering the ESC guidelines, we are not surprised to have much greater number of very high risk patients than high risk.

6. lines 227-230: were there no patients who switched from a NOAC to a VKA? And what was the proportion of patients with only a single recorded prescription for any OAC?

We appreciate this comment, which improved our manuscript.

We added the following in the results section:

Lines 235-241:“ In particular, 4133 patients (6.5% of the total) had a single recorded prescription for any anticoagulant: warfarin – 1496 (36.2%), rivaroxaban – 882 (21.34%), apixaban – 815 (19.71%), acenocoumarol – 397 (9.61%), dabigatran – 370 (8.95%), and edoxaban – 173 (4.19%). Furthermore, 3508 (5.5%) were prescribed VKA and switched to NOAC, 1504 (2.4%) were prescribed NOAC and switched to another NOAC, and 305 (0.5%) were prescribed NOAC and switched to VKA. The following patients were always prescribed the same NOAC: rivaroxaban – 8801 (22.18%), apixaban – 7052 (17.78%), dabigatran – 5219 (13.16%), and edoxaban – 782 (1.97%).”

VERSION 3 – REVIEW

REVIEWER	Eitaro Kodani Department of Cardiovascular Medicine, Nippon Medical School Tama Nagayama Hospital, Tokyo, Japan.
REVIEW RETURNED	21-Dec-2020

GENERAL COMMENTS	This revised manuscript by Pinto et al. focused on the current status of anticoagulation therapy in patients with atrial fibrillation (AF) in Northern Portugal using data of the AF-React study. Authors revised the manuscript according to the reviewers' comments. It appeared better. I have Minor comments; 1) Please standardize the decimal places of percentage throughout the manuscript. The, correct 19.7.8% and 13.9.8% in line 224.
--

REVIEWER	Tanja Mueller Strathclyde Institute of Pharmacy and Biomedical Sciences, University of Strathclyde, Glasgow, UK
REVIEW RETURNED	04-Jan-2021

GENERAL COMMENTS	Dear authors, your study is very interesting and relevant, and the manuscript is well written. In order to further improve clarity, I have two minor suggestions though: first, in line 149 (methods) you refer to the majority of the data being one-off data; it might be helpful to add what this refers to, if known (e.g. does the dataset contain the last available record for each variable/patient at the end of a calendar year?); and second, in line 247 (discussion) it says "the prevalence of AF was 3% in 2018"; I would recommend adding that this refers to the regional population aged 40 years or older. More importantly, though, I would strongly recommend to add to the limitations section that this study is, basically, a cross-sectional study; crucially, timing/sequencing of diagnoses and initiation of treatment are unknown - which might have implications for the interpretation of findings.
---

VERSION 3 – AUTHOR RESPONSE

Reviewer: 1

Dr. Eitaro Kodani, Nippon Medical School Tama Nagayama Hospital

Comments to the Author:

This revised manuscript by Pinto et al. focused on the current status of anticoagulation therapy in patients with atrial fibrillation (AF) in Northern Portugal using data of the AF-React study. Authors revised the manuscript according to the reviewers' comments. It appeared better. I have Minor comments;

Dear Reviewer,

Dr. Eitaro Kodani

We appreciate the new revision of the manuscript and all the comments you have made so far. They were very useful to improve the manuscript.

1) Please standardize the decimal places of percentage throughout the manuscript.

The, correct 19.7.8% and 13.9.8% in line 224.

We thank this correction. The percentages were corrected according to the Table 4. (line 228): "(17.5% vs 19.7%, 16.7% vs 13.0%, and 13.9% vs 13.1%, respectively)."

Reviewer: 2

Dr. Tanja Mueller, University of Strathclyde

Comments to the Author:

Dear authors,

your study is very interesting and relevant, and the manuscript is well written. In order to further improve clarity, I have two minor suggestions though:

Dear Reviewer,

Dr. Tanja Mueller

We appreciate the new revision of the manuscript and all the comments you have made so far. We appreciate these your minor revisions to clarify our manuscript.

first, in line 149 (methods) you refer to the majority of the data being one-off data; it might be helpful to add what this refers to, if known (e.g. does the dataset contain the last available record for each variable/patient at the end of a calendar year?); and second, in line 247 (discussion) it says "the prevalence of AF was 3% in 2018"; I would recommend adding that this refers to the regional population aged 40 years or older.

We appreciate your suggestion. In fact, it becomes more clear if we put the dataset contain the last available record for each variable/patient at the end of a calendar year:

(lines 152-154): "(age, gender, professional situation, ICPC-2 coded health problems, body mass index, cardiovascular risk score, and glomerular filtration rate)"

We appreciate your recommendation and add the suggested information:

(line 251): "in population aged 40 years or older."

More importantly, though, I would strongly recommend to add to the limitations section that this study is, basically, a cross-sectional study; crucially, timing/sequencing of diagnoses and initiation of treatment are unknown - which might have implications for the interpretation of findings.

We appreciate your suggestion and add this information in the discussion section:

(lines 329-331): "The fact that it is not known when treatment started for patients already diagnosed with AF before 2016 may be a limitation for interpreting the data. Besides that, the database didn't allow a detailed timing and sequencing of diagnosis, which might have implications for interpreting our findings."

And in the “Strengths and limitations of this study” section (lines 88-89):

- “The fact that it is not known when treatment started for patients already diagnosed with AF before 2016 may be a limitation for interpreting the data.”